# MiRNAs Action and Impact on Mitochondria Function, Metabolic Reprogramming and Chemoresistance of Cancer Cells: A Systematic Review

**DOI:** 10.3390/biomedicines11030693

**Published:** 2023-02-24

**Authors:** Daiane Rosolen, Emanuelle Nunes-Souza, Rafael Marchi, Maria Vitoria Tofolo, Valquíria C. Antunes, Fernanda C. B. Berti, Aline S. Fonseca, Luciane R. Cavalli

**Affiliations:** 1Research Institute Pelé Pequeno Príncipe, Faculdades Pequeno Príncipe, Curitiba 80230-020, PR, Brazil; 2Department of Oncology, Lombardi Comprehensive Cancer Center, Georgetown University, Washington, WA 20057, USA

**Keywords:** microRNA, mitomiRs, mitochondria, metabolic reprogramming, cancer, chemoresistance

## Abstract

MicroRNAs (miRNAs) are involved in the regulation of mitochondrial function and homeostasis, and in the modulation of cell metabolism, by targeting known oncogenes and tumor suppressor genes of metabolic-related signaling pathways involved in the hallmarks of cancer. This systematic review focuses on articles describing the role, association, and/or involvement of miRNAs in regulating the mitochondrial function and metabolic reprogramming of cancer cells. Following the PRISMA guidelines, the articles reviewed were published from January 2010 to September 2022, with the search terms “mitochondrial microRNA” and its synonyms (mitochondrial microRNA, mitochondrial miRNA, mito microRNA, or mitomiR), “reprogramming metabolism,” and “cancer” in the title or abstract). Thirty-six original research articles were selected, revealing 51 miRNAs with altered expression in 12 cancers: bladder, breast, cervical, colon, colorectal, liver, lung, melanoma, osteosarcoma, pancreatic, prostate, and tongue. The actions of miRNAs and their corresponding target genes have been reported mainly in cell metabolic processes, mitochondrial dynamics, mitophagy, apoptosis, redox signaling, and resistance to chemotherapeutic agents. Altogether, these studies support the role of miRNAs in the metabolic reprogramming hallmark of cancer cells and highlight their potential as predictive molecular markers of treatment response and/or targets that can be used for therapeutic intervention.

## 1. Introduction

MicroRNAs (miRNAs) are a class of small, highly conserved endogenous non-coding RNAs (19–25 nucleotides) that regulate gene expression post-transcriptionally. This regulation occurs mainly through binding to mRNA targets’ complementary sequences in the 3′ UTR region, blocking translation, and/or leading to mRNA degradation or destabilization. Although less common, miRNAs can activate gene expression by interacting with the 5′ UTR and gene promoter regions [1,2,3]. MiRNAs are involved in several mechanisms of tumor development and progression, acting as both oncogenes and tumor suppressors, depending on the type of cell and tissue [4,5]. In addition, a given miRNA can regulate multiple targets, and a single target can be regulated by multiple miRNAs, showing the intricate and complex interaction between miRNA and mRNA pairings [6].

MiRNAs regulate mitochondrial functions and homeostasis, such as metabolic reprogramming, redox signaling, mitochondrial membrane potential, calcium transport, mitochondrial fusion, fission dynamics, mitophagy, and apoptosis [7,8,9]. Metabolic reprogramming, also known as “deregulating cellular metabolism,” is one of the emerging hallmarks of cancer that occurs as a result of the metabolic plasticity of cancer cells [10,11]. In this metabolic “switch,” known as the “Warburg Effect,” cancer cells rely on glycolysis for their energy production, even in the presence of oxygen and normal functional mitochondria [12,13]. This switch increases glucose uptake and CO_2_ concentration for anabolic processes, that is, the biosynthesis of proteins, lipids, and nucleotides, needed to support cell proliferation and acidification of the microenvironment by releasing lactic acid, facilitating cancer cell invasion [14]. In addition, cancer cells take up large concentrations of glutamine, a non-essential aminoacid that acts in the carbon metabolism. The metabolism of glutamine reduces carbons to the tricarboxyic acid (TCA) cycle and generates several biosyntetic precursors and eletrons to oxidative phosporilation (OXPHOS [15]. Glutamine also acts in the nitrogen metabolism, and is required for the biosynthesis of amino acids, nucleotides, and amino sugars [16,17]. Other metabolic pathways impacted by glutamine include the nicotinamide adenine dinucleotide phosphate (NADPH) metabolism, which provides eletrons for anabolic reactions and redox balance [18,19]. NADPH production can occur through the serine-dependent one-carbon and pentose phosphate pathways. These pathways are involved in the production of oncometabolites, such as 2-hyroxyglutarate (2-HG), succinate, and fumarate, which activate NADPH and impair the formation of succinate dehydrogenase (SDH), fumarate hydratase (FH), and isocitrate dehydrogenase 1 or 2 (IDH1 or IDH2), which are important enzymes to TCA cycle process [18].

Under conditions of metabolic stress, particularly during deprivation of glucose, glutamine and oxygen, the scavenging of extracellular molecules, such as lipids, is an important mechanism to maintain cancer cells viability. The process of scavenging, rather than synthesizing, lipids supply the carbon metabolism [19,20]. Other mechanisms include fatty acid oxidation as energy supply and the activation of autophagy, which can eliminate damaged mitochondrial cells and their macromolecular components, and provide intracellular nutrients for cell survival and growth [20].

Therefore, it is clear that cancer cells can rely on more than one metabolic pathway to maintain their survival, and tumors often contain “energy-generating pathways,” subpopulations of cancer cells. In addition, energy reprogramming can also occur depending on the stage of tumor progression and the type of tissue and microenvironment [21,22]. Therefore, functional mitochondria are essential for tumor growth, not only for energy production but also for the biosynthesis of metabolites necessary for tumor proliferation, control of the production and release of reactive oxygen species (ROS), ion transport (calcium) homeostasis, and cell death [21,23].

The modulation of these mitochondrial-associated functions by miRNAs can occur by targeting the mRNAs from the cytoplasm (synthesized in the nucleus) and/or by importing miRNAs into the mitochondria to bind to mtDNA-encoded mRNAs [8,24,25,26,27]. This last process is suggested to involve the same proteins as nuclear miRNA biogenesis, such as AGO2 (argonaute 2) and PNPT1/PNPase (exoribonuclease polyribonucleotide nucleotidyl transferase) proteins, as well as other unknown miRNA-importing mediators [25,26,28]. In addition to the nucleus, miRNAs originate from the mitochondrial genome, also known as mitochondria-miRNAs (“mitomiRs”) [26,29,30,31]. Approximately 150 miRNA sequences have been identified in human mitochondrial DNA (mtDNA). However, the exact site of miRNA transcriptions originating from mitochondrial genes remains to be elucidated [32,33,34,35]. Recent studies in several tumor tissues have identified mitomiRs, showing their role in regulating critical cellular processes such as apoptosis, cell cycle, and cell metabolism [36,37].

MiRNAs are also involved in resistance to multiple chemotherapeutic agents and tumor recurrence in several types of cancer where mitochondria-mediated apoptotic pathways (and mitophagy) play essential roles [38,39]. Manipulating miRNA expression levels to directly reverse the impaired mitochondrial functions in cancer cells is a promising therapeutic strategy that could lead to increased sensitivity to treatment and reduced recurrence rates [40].

In this systematic review, we consider the emerging and prominent role of miRNAs, as they act as oncogenes or tumor suppressors in the regulation of the reprogramming metabolic hallmark of cancer. The main goal was to focus on searching for articles that described the role, association, and/or involvement of miRNAs (and mitomiRs) and their corresponding mRNA targets in regulating mitochondrial function, homeostasis, and metabolic reprogramming. The selected thirty-six articles reported on miRNAs regulating target genes involved in the aforementioned metabolic processes. In addition, some of these miRNAs have been reported to affect tumor resistance by mediating metabolic reprogramming and mitochondria-associated functions, which can point to a new perspective on cancer treatment based on cell metabolism.

## 2. Method

This review followed the Preferred Reporting Items for Systematic Reviews and Meta-Analyses (PRISMA) guidelines [41,42]. The review protocol was registered at the International Prospective Register of Systematic Reviews (PROSPERO) database under the identifier CRD42022319233.

### 2.1. Data Sources and Search Strategy

The databases Pubmed, Scielo, Lilacs, EMBASE, and Scopus, were searched using the terms “mitochondrial microRNA” and its synonyms (mitochondrial microRNA, mitochondrial miRNA, mito microRNA, or mitomiR), “reprogramming metabolism,” and “cancer” in the title or abstract. The articles searched were published from January 2010 to January 2022. A new search was conducted from January to September 2022 and included one article. Two reviewers independently performed the searches similarly for all of the databases. The online tool Rayyan (https://www.rayyan.ai/ (last accessed on 3 October 2022)) was used to analyze the selected studies. Duplicate articles were excluded, and two reviewers screened the remaining articles based on their title and abstract. Conflicting articles were evaluated by a third reviewer, followed by assessing the full text for relevance and eligibility.

### 2.2. Study Selection and Eligibility Criteria

Two reviewers independently evaluated and selected the studies according to the following inclusion and exclusion criteria. Inclusion criteria: (1) articles reporting the action of miRNAs in mitochondrial activities and the potential role of miRNAs on cellular mechanisms and pathways associated with mitochondrial functioning in cancer, including chemoresistance; (2) peer-reviewed articles written in English. Exclusion criteria: (1) articles on miRNA that did not report on the role, function, association, and/or involvement of mitochondria and cancer; (2) articles on miRNA analysis performed in animals; (3) non-original articles (reviews), editorials, letters from editors, book chapters, unpublished or non-peer-reviewed studies; (4) articles for which the full text was not available.

### 2.3. Data Extraction

After the selection and eligibility assessment of the studies, two reviewers extracted the following information independently: year of publication, name of first author, country, title, study aim, methodology type (in miRNA and/or mitochondria-related activity or function), sample source (experimental data on patients’ samples and/or cell lines, and/or data procured from online databases) used, miRNAs description, type of cancer analyzed, description of the main results, and conclusions.

### 2.4. Quality and Bias Evaluation

The Quality in Prognosis Studies (QUIPS) tool was used to assess the quality of the studies and the risk of bias, evaluating the studies in the following six categories: study participation, study attrition, prognostic factor measurement, outcome measurement, study confounding, and statistical analysis and reporting [39]. The articles were evaluated for quality according to the following classification—high quality (+): with little or no risk of bias; acceptable (+/−): moderate risk of bias; low quality (−): with a high risk of bias; and unsure (?). Based on this classification, the articles received a general evaluation as low, moderate, or high risk of bias.

## 3. Results and Discussion

### 3.1. Thirty-Six Articles Were Identified and Selected Following the PRISMA Guidelines

We identified 46 articles in the five selected databases using the selected search strategy and study eligibility criteria. We also manually included an additional 75 articles on the subject to increase the robustness of the present review. After removing duplicates, 97 articles were screened for relevance and compliance based on the inclusion and exclusion criteria. This analysis yielded 36 full-text articles for qualitative synthesis (Figure 1).

### 3.2. Most of the Articles Selected Presented a Low Risk of Bias

The risk of bias was determined for all of the studies using the six categories of the QUIPS tool [43]. The overall assessment of the six categories resulted in 27 studies with a low risk of bias and nine with a moderate risk of bias. Following this evaluation, all 36 studies were retained for further analysis.

### 3.3. Fifty-One miRNAs Were Described in 12 Types of Cancer

The 36 selected original articles for this systematic review were published between December 2010 and May 2022. The studies were published in eight countries: China (n = 20), India (n = 6), the United States (n = 4), Korea (n = 2), France, Japan, Mexico, and Norway (n = 1 each). The main aim of these 12 studies [38,44,45,46,47,48,49,50,51,52,53,54] was to comprehensively analyze the metabolic balance and/or chemoresistance associated with miRNAs in different cancers. In nine studies [55,56,57,58,59,60,61,62,63], the aim was to investigate the effect of miRNAs on mitochondrial function in cancer tissues and cell lines. Five studies [64,65,66,67,68] reported an the association between miRNAs and the regulation of glucose metabolism. Two studies [25,69] focused on the differential expression of miRNAs in tumor tissues and in the Hela cell line, and their enriched expression in mitochondria (mitomiRs). Another three articles [70,71,72] described the action of miRNAs in the hallmarks of cancer. Only two studies [73,74] associated miRNAs with the regulation of mitochondrial antioxidant enzymes, and three studies [75,76,77] described miRNAs as associated with apoptosis.

In total, 51 miRNAs were differentially expressed in 12 cancers: bladder (1 study), breast (24 studies), cervical (2 studies), colon (3 studies), colorectal (1 study), liver (1 study), lung (1 study), melanoma (2 studies), osteosarcoma (1 study), pancreatic (1 study), prostate (2 studies), and tongue (2 studies). High expression of miRNAs (n = 37) was observed in 17 studies [25,38,50,51,54,57,59,61,62,63,66,67,68,69,70,72,76] and lower expression (n = 12) in 9 studies [44,56,58,64,65,67,73,74,76]. Of the studies describing the overexpression of miRNAs, four involved the miR-181 family, reported in three tumor types: cervical [69], colon [54,62], and liver [68]. Additionally, four miRNAs, miR-21 [54,69], miR-34a-5p [63], miR-195 [57,70], and miR-210 [52,67], were analyzed in seven independent studies in several types of cancer, including cervical, lung, osteosarcoma, breast, and colon cancer. In studies with lower miRNA expression, miR-17 [73,74] and miR-125b [46,47] were observed in two independent studies, on prostate cancer and breast cancer, respectively. Two miRNAs, miR-let-7 [44,69] and miR-27a [45,51], were described to have opposite functions in four independent studies of breast cancer.

The miRNAs described in the 36 selected studies are presented in Table 1 and Table 2 according to their mode of action (as oncomiRs or tumor suppressors). The specific metabolic pathways and corresponding enzyme precursors that are targeted by the described miRNAs are presented in Table 3. Finally, miRNAs specifically associated with chemoresistance, and thus with potential value as predictive therapeutic markers and/or druggable targets, are presented in Table 4.

### 3.4. Thirty-One miRNAs Were Described with Oncogenic Mode of Action Mediating Metabolic Reprogramming and Mitochondria Related Functions

Thirteen of the 36 selected studies described miRNAs as having an oncogeniccompatible mode of action. These studies identified 31 miRNAs, of which the following 10 miRNAs were described with corresponding effects on the expression of target genes and/or cell function: miR-98 [50], miR-27a [51], miR-137 [61], miR-155 [66], miR-210 [52], miR-210-3p [67], miR-181c [62], miR-181a-5p [68], miR-593-5p [53], and miR-2392 [38].

The most frequently studied cancer types were breast [51,61,66,67], cervical [25,69], colon [52,62], and tongue [38,53], followed by bladder [50], colorectal [54], and liver [68] cancers (Table 1).

The miRNAs that acted as oncogenes were associated with different functions and regulated specific target genes involved in several cancer phenotypes (except the studies [25,54,69]). The most frequent alterations observed in the gene expression deregulation of these miRNAs were drug resistance, cell metabolism (lactic acid secretion and OXPHOS), apoptosis, colony formation, cell growth and cell cycle, and development of metastasis [51].

Breast cancer was the most frequently studied cancer, with reported alterations in the expressions of miR-27a [51], miR-137 [61], miR-155 [66], and miR-210-3p [67]. In a study by Zhou et al. (2015), overexpression of miR-27-a was associated with downregulated expression of the BCL2 antagonist/killer 1 (*BAK*) gene and the second mitochondria-derived activator of caspase/DIABLO-IAP binding mitochondria protein/X-linked inhibitor of apoptosis (SMAC/DIABLO/XIAP) axis, resulting in a reduction in the apoptosis and chemosensitivity of cancer cells. It also increased tumorigenicity, as observed by the increase in colony formation and metastasis development [51]. Hu et al. (2020) also demonstrated a reduction in apoptosis with miR-137 overexpression by downregulating the FUN14 domain containing one (*FUNDC1*) gene. In this study, miR-137 overexpression led to a decrease in ROS levels [61]. Overexpression of miR-155 was shown to distinctly affect the FOXO3a/c-MYC axis and promote tumor growth by increasing glucose uptake and glycolysis [66]. Finally, in a breast cancer study by Du et al. (2020), overexpression of miR-210-5p led to the downregulation of glycerol-3-phosphate dehydrogenase 1-like (*GPD1L*) and cytoglobin (*CYGB*) target genes, causing metabolic alterations in the cells, with an increase in glucose and lactate uptake and a reduction in apoptosis [67]. Alterations in the expression of miR-210 were reported in colon [52] and colorectal cancer [54] studies. In colon cancer, its overexpression led to the downregulation of iron-sulfur cluster assembly enzyme (*ISCU*) and cytochrome C oxidase assembly factor heme A (*COX10*) genes, and increased cell survival in hypoxic microenvironment [52].

In other tumor types, the most cited miRNAs in the selected studies were those from the miR-181 (a, b, and c) family, reported in cervical [69], colon [62], colorectal [54], and liver [68] cancers. In cervical cancer, miR-181b was observed with high expression in the mitochondria of the HeLa cells [69]. In the nuclear factor (erythroid-derived 2)-like 2 (NFE2L2/NRF2) knockdown colon cancer study by Jung et al. (2017), significant mitochondrial dysfunction was reported with miR-181c overexpression. This altered expression led to the downregulation of mitochondria-encoded cytochrome c oxidase subunit-1 (mt-CO1), and these changes induced adenosine monophosphate (AMP)-activated protein kinase-a (AMPKa) activation and its subsequent metabolic adaptation signaling, including a reduction in OXPHOS. In a colorectal cancer study, miR-181 overexpression led to the transformation of precancerous cells in adenocarcinomas [54]. Finally, in a liver cancer study [68], overexpression of miR-181a-5p was shown to cause electron transport chain (ETC) remodeling, which reduced OXPHOS and increased cell survival in a hypoxic microenvironment, as well as glucose consumption and lactic secretion.

In two selected studies of tongue cancer [38,53], overexpression of miR-593-5p [53] and miR-2392 [38] affected the expression of their targets, mitochondrial fission factor (MFF) and argonaute 2 (AGO2), respectively, and led to an increase in chemoresistance. In addition, miR-593-5p led to mitochondria fission, and miR-2392 led to a reduction in OXPHOS and increased glycolysis.

Altogether, these studies demonstrate the prominent role of miRNAs in the cell metabolism and reprogramming of cancer cells by regulating critical mRNA targets of both glycolysis- and mitochondrial-mediated pathways (Figure 2). They also demonstrate that miRNAs with the same mode of action can affect these pathways by regulating distinct targets, which highlights their versatile regulation of gene expression**.**

A summary of the studies above, distributed per tumor type with the identified miRNAs, their corresponding target gene mechanism(s) of action, and the impact on cancer cell phenotypes, is presented in Table 1.

### 3.5. Twenty-One miRNAs Were Described with Suppressive Mode of Action Mediating Metabolic Reprogramming and Mitochondria Related Functions

Twenty-six studies conducted during the search period reported 21 miRNAs with tumor suppressive actions: miR-let-7a [44,55], miR-1 [56], miR-17* [73], miR-17-3p [74], miR-24-2 [77], miR-27a [45], miR-34a-5p [63], miR-125b [46,47], miR-128 [64], miR-133a [48], miR-140-5p [65], miR-195 [57,70,77], miR-200a [58], miR-223 [75], miR-340 [71], miR-342-3p [49], miR-365-2 [77], miR-519d [76], miR-663 [59], miR-1291 [72], and miR-4485 [60]. Of these studies, 20 were in breast cancer, 1 was in lung cancer and osteosarcoma, 2 were in melanoma, 2 were in prostate cancer, and 1 was in pancreatic cancer (Table 2).

The expression of the altered miRNAs varied based on the main mRNA targets and tumor type. The most reported impact of tumor suppressive action on cancer phenotypes was on cell proliferation, apoptosis, and cytotoxicity to chemotherapeutic agents. Others affected the cell metabolism processes, such as glycolysis and mitochondrial organization, structure, and function [56,72,77].

The miRNAs that were most frequently involved in the breast cancer studies were: miR-let-7a [44,55], miR-125b [46,47], and miR-195 [57,70,77]. MiR-let-7a was reported to regulate the expression of distinct mRNA targets, including aminoadipate-semialdehyde dehydrogenase-phosphopantetheinyl transferase (*AASDHPPT*), BTB domain and CNC homolog 1 (*BACH1*), fatty acid synthase (*FASN*), glucose-6-phosphate dehydrogenase (*G6PD*), heme oxygenase 1 (*HMOX1*), inosine monophosphate dehydrogenase 2 (*IMDH2*), NADH-ubiquinone oxidoreductase chain 4 (*ND4*), and stearoyl-CoA desaturase (*SCD*) [44,55]. Sharma et al. (2021) reported that the altered expression of miR-let-7a was associated with an increase in lactate concentration and a decrease in adipogenesis, through the regulation of the *ND4* target gene and reduction of OXPHOS [55]. Furthermore, in a study by Serguienko et al. (2015), the suppressive action of miR-let-7a was associated with a decrease in tumor cell proliferation and an increase in the sensitivity of tumor cells to chemotherapy. This study also demonstrated the overexpression of miR-let-7a in melanoma cells, which regulated the targets *AASDHPPT*, *BACH1*, *FASN*, *G6PD*, *IMPDH2*, and *SCD*, and led to a decrease in cell proliferation and higher responsiveness to chemotherapy [44].

MiR-125b interacted with two targets, HCLS-1-associated protein X-1 (*HAX-1*) and myeloid-cell leukemia 1 (*MCL-1*). In a study by Hu et al. (2018), overexpression of miR-125b was associated with reduced HAX-1 expression in breast cancer cells exposed to doxorubicin. This expression increased caspase 1 and ROS activity, resulting in increased cell death (apoptosis), chemosensitivity, and mitochondrial damage [46]. Other breast cancer studies have demonstrated a similar impact via the MCL-1 gene. The overexpression of miR-125b reduced the expression of MCL-1, increasing caspase-3 and apoptosis and reducing doxorubicin resistance [47].

Finally, miR-195 was upregulated in three studies of breast cancer [57,70,77]. In a study by Singh et al. (2011), upregulation was associated with increased apoptosis and a decrease in BCL2 Apoptosis Regulator (*BCL2*) expression [77]. In another study by Singh et al. (2015), miR-195 overexpression was observed to affect the expression of acetyl-CoA carboxylase alpha (*ACACA*), cytochrome P450 family 27 subfamily B member 1 (*CYP27B1*), *FASN*, and 3-hydroxy-3-methylglutaryl-CoA reductase (*HMGCR*) [70]. Finally, in a study by Purohit et al. (2019), miR-195 was associated with alterations in mitochondrial dynamics and homeostasis and an increase in apoptosis by downregulating the expression of the mitofusin-2 (*MFN2*) gene [57].

The MiRNAs that were less frequently altered in breast cancer studies were miR-24-2 [77], miR-27a [45], miR-128 [64], miR-133a [48], miR-140-5p [65], miR-340 [71], miR-342-3p [49], miR-519d [76], miR-663 [59], and miR-4485 [60]. In a study by Singh et al. (2012), the overexpression of miR-24, miR-195, and miR-3652 resulted in the downregulation of *BCL2*, which altered the mitochondrial membrane potential, increased the release of cytochrome C in the cytoplasm, and triggered apoptosis [77]. MiR-27a, on the other hand, acted alone, affecting the expression of its targets cystathionine gamma-lyase (*CTH*), NFE2-like BZIP transcription factor 2 (*NFE2L2*), and solute carrier family 7 member 11 (*SLC7A11*), leading to an increase in ROS and reducing autophagy and the chemoresistance of breast cancer cells [45]. The overexpression of miR-128, miR-340, and miR-342-3p was reported to affect cell metabolism by altering the expression of the insulin receptor (*INSR*) and insulin receptor substrate 1 (*ISR1*), mitochondrial calcium uniporter (*MCU*), and monocarboxylate transporter 1 (*MCT-1*) targets genes, respectively [49,64,71].

In a study on breast cancer and melanoma by Zhang et al. (2019), miR-1 was up-regulated, affecting the expression of ATP synthase membrane subunit 6 (*ATP6*), cytochrome C oxidase subunit 1 (*COX1*), glycerol-3-phosphate dehydrogenase 2 (*GPD2*), mitochondrial inner membrane organizing system 1 (*MINOS1*), NADH dehydrogenase subunit 1 (*ND1*), and *ND4.* These alterations decreased tumorigenicity and caused disorganization of the mitochondrial crest [56].

Yi et al. (2022) showed that the overexpression of miR-34a-5p led to downregulation of the mitochondrial inner membrane protein MPV17-like 2 (MPV17L2) in lung cancer and osteosarcoma cell lines. The miR-34a-5p suppressed the expression of MPV17L2, resulting in lower levels of respiratory chain complex I activities and intracellular ATP, a significant decrease in mitochondrial NADH dehydrogenase 1 (MT-ND1) protein levels, and an increase in oxidative stress, resulting in elevated apoptotic cell death. [63].

Two prostate cancer studies [73,74] reported alterations in the expression of miR-17* and miR-17-3p. In a study by Xu et al. (2010), the high expression of miR-17* increased mitochondrial ROS, which resulted in increased cytotoxicity to disulfiran in the cells and, consequently, cell death [73]. Xu et al. (2018) reported that the overexpression of miR-17-3p was positively associated with ionizing radiation, increasing the radiosensitivity and cell death of prostate tumor cells. In both studies, changes in miR-17 expression occurred via glutathione-dependent peroxidase (*GPX2*), manganese superoxide dismutase (*MnSOD*), and thioredoxin reductase 2 (*TRXR2*) targets expression [74].

Finally, a study by Chen et al. (2020) in pancreatic and breast cancer showed the overexpression of miR-1291, which acts in the estrogen-related receptor alpha (ERRα) and carnitine palmitoyl transferase 1C–CPT1C (ERRα-CPT1C axis). This miRNA alteration led to mitochondrial dysfunction and decreased cell metabolism, proliferation, invasion, and tumorigenesis [72].

These studies indicated the putative tumor suppressive action of the described miRNAs on metabolic reprogramming and mitochondria-related functions, highlighting the need for further evidence for their potential application in cancer pharmacological therapy. Interestingly, the mitochondrial action of the same described miRNAs can also occur in other human diseases, such as cardiac diseases, supporting the discovery of new treatments based on epigenetic targets [78]. Nonetheless, these results show the diverse and complex regulatory action of miRNAs in the metabolic processes, by regulating interactions among multiple enzymes and complex metabolic components, which are among the major challenges for their clinical application.

The summary of these studies per tumor type with the identified tumor suppressor miRNAs and their corresponding target genes, mechanisms of action, and impacts on cancer cell phenotypes are presented in Table 2. The involvement of these miRNAs in the distinct cell metabolic process is shown in Figure 2, and the specific metabolic pathways and corresponding enzyme precursors that are targeted by these miRNAs (identified using https://www.proteinatlas.org (last accessed on 25 January 2023) (metabolic search)) are presented in Table 3.

### 3.6. Nine miRNAs Were Identified Acting on Tumor Chemoresistance Mediating Metabolic Reprogramming and Mitochondria Related Functions

Chemoresistance is one of the main problems in cancer treatment and can cause a lack of treatment response, tumor recurrence, and high mortality rates [39]. MiRNAs play a key role in chemoresistance by regulating target genes involved in diverse cellular mechanisms, including metabolic reprogramming [43,61,79].

Of the studies above, which describe the role of miRNAs as oncomiRs and/or tumor suppressors, 11 of them specifically reported their association with chemoresistance (Table 4). Nine miRNAs were described in these studies: miR-98, miR-27a, miR-125b, miR-133a, let-7a, miR-223, miR-519d, miR-593-5p, and miR-2392. Most chemoresistance-associated miRNAs were reported in breast cancer [44,45,46,47,48,51,75,76], followed by tongue cancer [38,53], melanoma [44], and bladder cancer [50].

Luan et al. (2018) reported that, in bladder cancer, the overexpression of miR-98 decreased *LASS2* (LAG1 longevity assurance homolog 2) expression, leading to a decrease in mitochondrial fusion and an increase in the mitochondrial membrane potential which conferred resistance to cisplatin and doxorubicin [50].

For breast cancer, two studies showed an association between miR-27a and cytotoxicity to cisplatin, doxorubicin, and/or paclitaxel [45,51]. In a study by Ueda et al. (2020), miR-27a was described as having a tumor suppressive action, considering that its overexpression increased mitochondrial ROS and rendered MCF-7 and MDA-MB-231 cells more sensitive to doxorubicin and paclitaxel by inhibiting *CTH* (Cystathionine gamma-lyase), *xCT* (Cystine/glutamate transporter), and *NRF2* (Nuclear factor erythroid-derived 2-like) expression [45]. In a study conducted by Zhou et al. (2015), miR-27a presented an oncogenic function, with overexpression associated with the inhibition of the *BAK* (*BCL2* family member) and SMAC/DIABLO/XIAP pathways, increasing resistance to cisplatin in T-47D breast cancer cells [51].

MiR-125b was described in two breast cancer articles to be associated with sensitivity to doxorubicin [46,47]. Hu et al. (2018) showed that the overexpression of miR-125b increased sensitivity to doxorubicin in MCF-7 cells resistant to doxorubicin (MCF-7/DOX R), which was mediated by the downregulation of the *HAX-1* gene and increased Caspase 9 and ROS levels [46]. Using a different in vitro model, Xie et al. (2015) demonstrated that inhibition of miR-125b decreased the sensitivity of tumor cells to doxorubicin by increasing the expression of its target *MCL-1* [47]. However, Yuan et al. (2015), using the same MCF-7/DOX R model above, reported the involvement of a different miRNA and target gene modulating the doxorubicin cytotoxicity; the decreased expression of miR-133a increased doxorubicin sensitivity by increasing the expression of the target Uncoupling Protein 2 (*UCP-2*) [48].

MiR-let-7a [44], miR-223 [75], and miR-519d [76] have also been reported to impact the resistance to chemotherapy of breast cancer cells. A study conducted by Serguienko et al. (2015) showed that the overexpression of miR-let7a inhibited the expression of *BACH1*, *G6PD*, *IMPDH2*, *FASN*, *SCD*, *AASDHPPT*, and *ND4*, and, consequently, increased the mitochondrial ROS and chemosensitivity of MDA-MB-231 triple negative breast cancer (TNBC) cells to doxorubicin. The same was observed in WM239 metastatic melanoma cells [44]. In a study by Sun et al. (2016), the induction of miR-223 expression in TNBC stem cells increased their sensitivity and cytotoxicity to doxorubicin or cisplatin, mediated by the decrease in *HAX* expression and increase in mitochondrial ROS [75]. Another study in breast cancer stem cells reported that overexpression of miR-519d in cells incubated with cisplatin decreased *MCL-1* expression and increased cytochrome C, activating the SMAC/DIABLO pathway and leading to apoptosis [76].

In the tongue cancer studies of Fan et al. (2015; 2019), miR-593-5p and miR-2392 were associated with cisplatin resistance [38,53]. In a study by Fan et al. (2019), conducted in CAL-27 and SCC-9 oral squamous carcinoma cells, overexpressed miR-2392 co-immunoprecipitated *AGO2* which, in turn, decreased OXPHOS and increased glycolysis, making the cells more resistant to cisplatin [38]. In another study conducted by Fan et al. (2015) in the same cell models, the overexpression of miR-593-5p with the overexpression of breast cancer gene 1 (*BRCA1*) decreased *MFF* expression, conferring cisplatin resistance to the cells [53].

These studies (summarized in Table 4 and Figure 3) highlight the essential role of miRNAs in conferring tumor resistance by modulating mitochondria-mediated cell processes. They also point to miRNAs as having potential use as predictive molecular markers of treatment response and/or as molecular targets for therapeutic intervention. Nonetheless, additional in vitro studies in well-established drug resistant cell models and/or tumor cells that are directly immortalized from patients’ tumors are required.

## 4. Conclusions

In conclusion, based on the 36 studies identified, this systematic review compiles evidence of the involvement of miRNAs and their corresponding mechanisms of action and biological impact in the metabolic reprogramming of cancer cells. By regulating target genes of diverse cancer-associated signaling pathways, miRNAs have been reported to be involved in cell metabolic processes, mitochondrial dynamics, mitophagy, apoptosis, redox signaling, and resistance to chemotherapeutic agents. As increasing evidence has emerged regarding the role of miRNAs in metabolic reprogramming and other associated hallmarks of cancer, their potential as predictive molecular markers of treatment response and/or druggable targets can be determined.

## Figures and Tables

**Figure 1 biomedicines-11-00693-f001:**
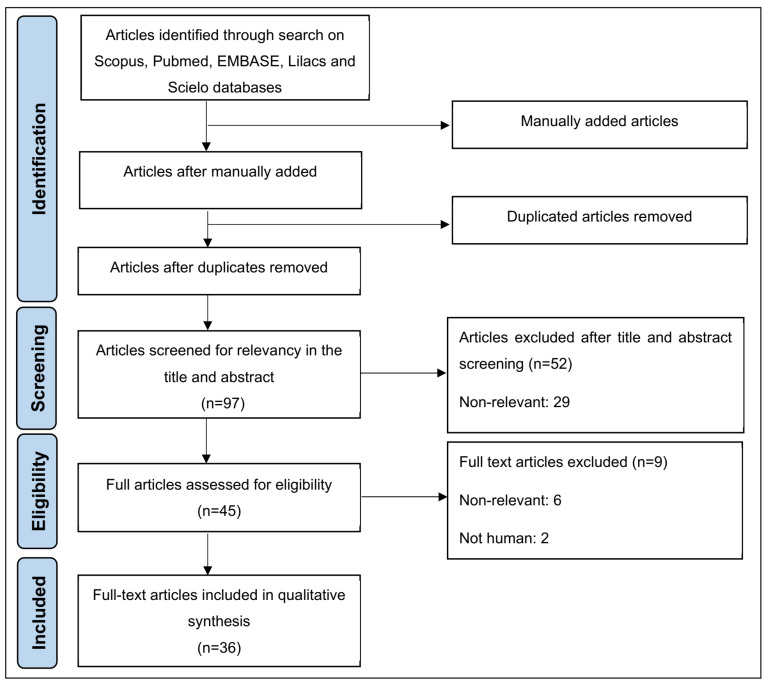
Flow diagram of the identification and selection of the studies, following the methodological steps of the PRISMA guidelines.

**Figure 2 biomedicines-11-00693-f002:**
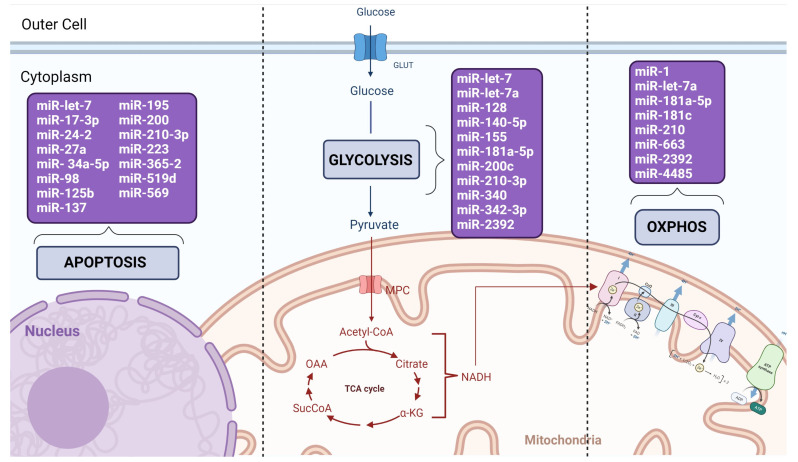
Overview of the involvement of the identified miRNAs in the apoptosis and metabolic processes of the cancer cells. Fifteen miRNAs were identified which suppress or induce apoptosis by regulating the expression of pro-and anti-apoptotic gene-targets. Eleven miRNAs were observed affecting specific steps of glycolysis, from the glucose uptake to pyruvate synthesis, by regulating several glycolytic enzymes. Finally, eight miRNAs directly affected OXPHOS by regulating gene-targets involved in mitochondria function and homeostasis.

**Figure 3 biomedicines-11-00693-f003:**
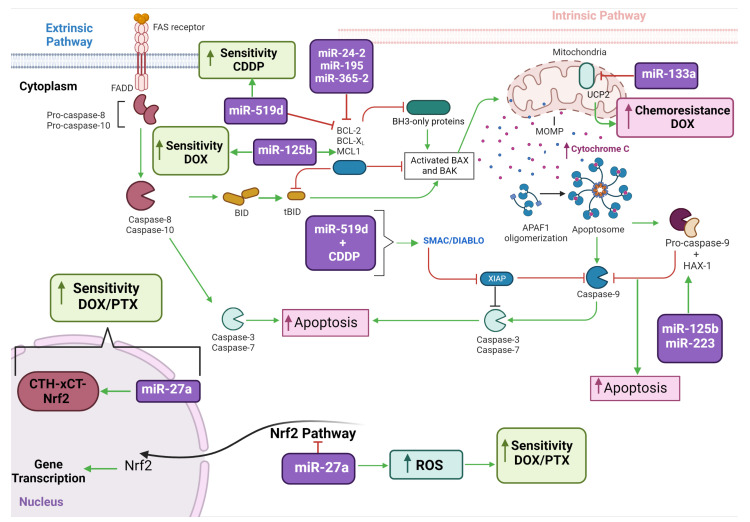
MiRNAs described in this systematic review with potential use as predictive molecular markers of treatment response and/or druggable targets. The figure illustrates the distinct mechanisms of the extrinsic and intrinsic pathways of apoptosis, by which the miRNAs mediate cytotoxicity to commonly-used cancer chemotherapeutic agents.

**Table 1 biomedicines-11-00693-t001:** MiRNAs identified with oncogenic function, their target genes, mechanism of action, and corresponding impact on distinct cancer cells phenotypes (presented by type of cancer).

MiRNA	Target Gene	Cancer Type	Mechanism of Action	Biological Impact	Reference
miR-98	*LASS2*	Bladder cancer	↑miR-98 ↓LASS2→↓Mitochondrial fusion ↑Mitochondrial membrane potential	↓Apoptosis ↑Chemoresistance	[50]
miR-27a	*BAK-SMAC/DIABLO/XIAP*	Breast cancer	↑miR-27a→↓BAK→↓SMAC/DIABLO/XIAP	↑Colony formation ↓Chemosensitivity ↑Metastasis ↓ Apoptosis	[51]
miR-137	*FUNDCc1*	Breast cancer	↑miR-137→ ↓FUNDC1→↓ROS	↓Apoptosis ↓Autophagy/Mitophagy ↑Cell cycle	[61]
miR-155	*PIK3R1-FOXO3a-cMYC* axis; *p85α-FOXO3a-cMYC*	Breast cancer	↑miR-155→↓PIK3R1-FOXO3a-cMYC ↑miR-155→↓*p85α-FOXO3a-cMYC*	↑Glucose and glycolysis ↑Tumor growth	[66]
miR-210-3p	*CYGB* *GPD1L* *HIF-1α* and *p53* activity via *GPD1L* and *CYGB*	Breast cancer	↑miR-210-3p ↓CYGB ↓GPD1L ↑HIF-1α↓p53→↑Aerobic glycolysis ↓Apoptosis	↑Colony formation, ↑Extracellular acidification rate ↑Glucose uptake ↑Lactate production ↓Serum starvation-induced cell apoptosis.	[67]
miR-let-7i miR-21 miR-23a miR-29a miR-30a miR-31 miR-181b miR-452	Not described	Cervical cell line (HeLa)	↑miR-let-7i ↑miR-21 ↑miR-23a ↑miR-29a ↑miR-30a ↑miR-31 ↑miR-181b ↑miR-452	Systematic analysis shows a significant overexpression of these miRNAs and their enrichment in the mitochondrial RNA fraction	[69]
miR-328 miR-494 miR-513a-5p miR-638 miR-1201 miR-1246 miR-1275 miR-1908 miR-1972	Not described	Cervical cell line (HeLa)	↑miR-328 ↑miR-494 ↑miR-513a-5p ↑miR-638 ↑miR-1201 ↑miR-1246 ↑miR-1275 ↑miR-1908 ↑miR-1972	Systematic analysis shows a significant overexpression of these miRNAs and their enrichment in the mitochondrial RNA fraction	[25]
miR-181c	*AMPKa* *mt-CO1*	Colon cancer	↑miR-181c ↓mt-CO1 ↑MMP disturbance and ETC dysfunction→↓levels of ATP ↑AMPKa	↓Mitochondrial OCR - OXPHOS (NFE2L2/NRF2- knockdown colon cancer cells)	[62]
miR-210	*COX10* *ISCU*	Colon cancer	↑ miR-210 ↓COX10/ISCU ↑ROS	↑Cell survival under hypoxic microenvironment	[52]
miR-21 miR-24 miR-181 miR-210 miR-378	Not described	Colorectal cancer	↑ miR-21 ↑miR-24 ↑miR-181 ↑miR-210 ↑miR-378	↑Transformation of precancerous polyps to CRC adenocarcinoma	[54]
miR-181a-5p	*mt-CO2* *mt-CYB*	Liver cancer	↑miR-181a-5p ↓mt-CYB ↓mt-CO2 →ETC remodeling →↓OXPHOS ↑*GLUT1* ↑*HK2*	↑Cell survival under hypoxic microenvironment ↑Glucose consumption ↑Lactic acid secretion ↑LDH	[68]
miR-593-5p	*MFF*	Tongue cancer	↑BRCA1→↑miR-593-5p→ ↓MFF	↓Cisplatin sensitivity ↑Mitochondrial fission	[53]
miR-2392	*AGO2*	Tongue cancer	↑miR-2392 ↑AGO2 →↓OXPHOS ↑Glycolysis	↑Chemoresistance ↑Glycolysis	[38]

Legend. “↑”: enhanced; “**↓**”: reduced; “→”:resulted in; **AGO2**: Argonaute 2; **AMPKα**: Adenosine monophosphate (AMP)-activated protein kinase-a; **ATP**: Adenosine triphosphate; ***BAK***: *BCL2* Antagonist/killer 1; ***BRCA1***: Breast cancer gene 1; ***COX10***: Cytochrome C Oxidase Assembly Factor Heme A:Farnesytransferase COX10; **CRC**: Colorectal cancer; ***CYGB***: Cytoglobin; ***DIABLO***: DIABLO-IAP binding mitochondria protein; **ETC**: Electron transport chain; ***FOXO3A***: transcription factor Forkhead box protein O3; ***FUNDC1***: FUN14 domain containing 1; ***GLUT1***: Glucose transporter type 1; ***GPD1L***: Dehydrogenase 1-like; ***HIF-1α***: Hypoxia-inducible factor 1-alpha ***HK2***: Hexokinase 2; **ISCU**: Iron-Sulfur Cluster Assembly Enzyme; ***LASS2***: LAG1 longevity assurance homolog 2; **MFF**: Mitochondrial fission factor; **MMP**: mitochondrial membrane potential; **mt-CO1**: Mitochondria-encoded cytochrome c oxidase subunit-1; **mt-CO2**: Mitochondrially Encoded Cytochrome C Oxidase II mt-CYB; **mt-CYB**: Mitochondrially Encoded Cytochrome B; ***MYC***: MYC Proto-Oncogene, BHLH Transcription Factor; ***NFE2L2/NRF2***: Nuclear factor (erythroid-derived 2)-like; ***OCR***: Oxygen consumption rate; **OXPHOS**: Oxidative phosphorylation; ***PIK3R1***: phosphoinositide-3-kinase regulatory subunit alpha; **p53:** Tumor protein 53**; p85α**: Modular protein binds and inhibits the enzymatic activity of class IA PI3K catalytic subunits; **ROS**: Reactive oxygen species; 2; ***SMAC***: Second mitochondria-derived activator of caspase; ***XIAP***: X-linked inhibitor of apoptosis.

**Table 2 biomedicines-11-00693-t002:** MiRNAs identified with tumor suppressor function, their target genes, mechanisms of action, and corresponding impacts on distinct cancer cells phenotypes (presented by type of cancer).

MiRNA	Target Gene	Cancer Type	Mechanism of Action	Biological Impact	Reference
let-7a	*ND4*	Breast cancer	↑miR-let-7a ↓ND4 → ↓OXPHOS	↓Adipogenesis ↑Lactate concentration (glycolysis)	[55]
let-7a	*AASDHPPT* *BACH1* *FASN* *G6PD* *HMOX1* *IMPDH2* *SCD*	Breast cancer and melanoma	↑miR-let-7a → ↓*AASDHPPT* ↓BACH1 ↓*FASN* ↓*G6PD* ↓*SCD* ↓IMPDH2 ↑HMOX1 → ↑*ROS* ↑OXPHOS and glycolysis	↑Chemosensitivity ↓Proliferation	[44]
miR-1	*ATP6* *COX1* *GPD2* *LRPPRC* *MINOS1* *ND1* *ND4*	Breast cancer and melanoma	↑miR-1 ↓ ATP6 ↓COX1, ↓ GPD2 ↓ LRPPRC ↓MINOS1 ↓ND1 ↓ ND4	↑Mitochondrial cristae organization and morphology ↓Tumorigenicity	[56]
miR-24-2 miR-195 miR-365-2	*BCL2*	Breast cancer	↑miR-24-2 ↑miR-195 ↑miR-365-2 ↓BCL2 ↓ΔΨm ↑cytochrome c protein (in the cytosol)	↑Apoptosis	[77]
miR-27a	*CTH* *NFE2L2* *SLC7A11*	Breast cancer	↑miR-27a ↓CTH ↓xCT ↓NRF2↑ROS	↓Autophagy ↓Chemoresistance	[45]
miR-125b	*HAX-1*	Breast cancer	↑ miR-125b + DOX → ↓HAX-1 ↑caspase-9 ↑ROS	↑Apoptosis ↑Cell death ↑ Chemosensitivity ↑Mitochondrial damage	[46]
miR-125b	*MCL-1*	Breast cancer	↑miR-125b ↓MCL-1→ ↑caspase-3	↑Apoptosis ↓Doxorubicin resistance	[47]
miR-128	*INSR* *IRS1*	Breast cancer	↑miR-128 ↓INSR ↓IRS1→↓mtDNA	↓Glucose metabolism (glycolysis) ↓Proliferation	[64]
miR-133a	*UCP-2*	Breast cancer	↑miR 133a → ↓UCP-2	↓Chemoresistance	[48]
miR-140-5p	*GLUT1*	Breast cancer	↑miR-140-5p ↓GLUT1→↓glycolysis ↑mitochondrial respiration	↓Proliferation (glycolysis)	[65]
miR-195	*ACACA* *CYP27B1* *FASN* *HMGCR*	Breast cancer	↑miR-195 ↓ ACACA →↓biosynthesis of fatty acid ↓ FASN ↓CYP27B1 → ↓mitochondrial calcium levels ↓HMGCR →↓mevalonate pathway	↓Fatty acid (via de novo lipogenesis)	[70]
miR-195	*MFN2*	Breast cancer	↑miR-195 ↓MFN2→↓mitochondrial dynamics ↓mitochondrial calcium homeostasis	↑Apoptosis	[57]
miR-200a	*TFAM*	Breast cancer	↑miR-200a ↓TFAM→↓mtDNA copy number	↓Cell proliferation	[58]
miR-223	*TRAIL/HAX-1*	Breast cancer	↑miR-223 ↓TRAIL → ↓HAX-1	↑Apoptosis ↑Chemosensitivity	[75]
miR-340	*MCU*	Breast cancer	↑miR-340 ↓MCU→↓glycolysis	↓Metastasis (glycolysis)	[71]
miR-342-3p	*MCT-1*	Breast cancer	↑miR-342-3p ↓ MCT-1 → ↓ extracellular lactate ↑glucose consumption	↓ Lactate transportation to inside cell	[49]
miR-519d	*MCL-1*	Breast cancer	↑miR-519d ↑cisplatin → ↓MCL-1 ↑cytochrome C ↑SMAC/DIABLO	↑Apoptosis ↑ Cell death ↑Chemosensitivity	[76]
miR-663	OXPHOS genes	Breast cancer	↑miR-663 ↑OXPHOS	↓Tumorigenesis	[59]
miR-1291	*CPT1C* *ERRα*	Breast cancer and pancreatic cancer	↑miR-1291 ↓*ERRα* → ↓*CPT1C*	↓Invasion ↑Mitochondrial dysfunction ↓ Proliferation ↓Tumorigenesis ↓ Cell metabolism	[72]
miR-4485	16S rRNA	Breast cancer	↑miR-4485 ↓ ΔΨm ↓16S rRNA →↓respiratory complex I →↑ROS	↑Cell death	[60]
miR-34a-5p	MPV17L2	Lung cancer and osteosarcoma	↓ND1 expression ↓mitochondrial respiration ↓ATP production ↑ROS accumulation	↑ Apoptotic cell death	[63]
miR-17*	*GPX2* *MnSOD* *TRXR2*	Prostate cancer	↑miR-17 ↓Gpx2 ↓MnSOD ↓TRXR2 → ↑ROS	↑Cytotoxicity ↑Cell death ↓Tumorigenicity	[73]
miR-17-3p	*GPX2* *MnSOD* *TRXR2*	Prostate cancer	↑miR-17-3p ↓Gpx2 ↓MnSOD ↓TRXR2 → ↑ROS ↓mitochondrial respiration	↑Cell death ↑Ionizing radiation ↑Radiosensitivity	[74]

Legend. “↑”: enhanced; “**↓**”: reduced; “→”: resulted in; **16S rRNA**: 16S ribossomal RNA; ***AASDHPPT***: 4-Phosphopantetheinyl transferase; ***ACACA***: Acetyl-CoA carboxylase; ***ATP6*:** ATP Synthase Membrane Subunit 6; ***BACH1***: BTB Domain And CNC Homolog 1; ***BCL2****:* B cell lymphoma 2; ***COX1***: Cytochrome C oxidase subunit 1; ***CPT1C***: Carnitine palmitoyltransferase 1C; ***CTH***: Cystathionine gamma-lyase; ***CYP27B1****:* subfamily B, polypeptide 1; ***DIABLO***: Direct IAP binding protein with Low pl; **DOX**: Doxorubicin; ***ERRα***: Estrogen-related receptor α; **FASN**: fatty acid synthase; ***G6PD***: Glucose-6-phosphate dehydrogenase**; *GLUT 1***: Glucose transporter 1; ***GPD2***: Glycerol-3-phosphate dehydrogenase 2; ***GPX2***: Glutathione peroxidase-2; **HAX-1**: hematopoietic cell-specific protein 1-associated protein X-1; ***HMGCR***: 3-Hydroxy-3-Methyl glutaryl CoA reductase; ***HMOX1***: Heme oxygenase 1; ***IMPDH2***: De novo guanine nucleotide biosynthesis; ***INSR***: Insulin receptor; ***IRS1***: Insulin receptor substrate 1; ***LRPPRC***: Leucine-rich pentatricopeptide-repeat containing; ***MCL-1***: Myeloid cell leukemia-1; **MCU**: Mitochondrial calcium uniporter; ***MCT1***: Monocarboxylate transporter 1; ***MFN2***: Mitofusin-2; ***MINOS1***: Mitochondrial inner membrane organizing system 1; ***MnSOD***: Manganese superoxide dismutase; ***MPV17L2:*** Mitochondrial inner membrane protein MPV17 like 2 mtDNA: Mitochondrial DNA; ***ND1***: NADH dehydrogenase 1; ***ND4***: NADH dehydrogenase 4; ***NFE2L2***: Nuclear factor erythroid-derived 2-like 2; **OXPHOS**: Oxidative phosphorylation; **ROS**: reactive oxygen species; ***SMAC***: Second mitochondria-derived activator of caspase; ***SCD***: Stearoyl-CoA desaturase; ***SLC7A11***: Solute carrier family 7 member 11; ***TFAM***: Mitochondrial transcription factor A; ***TRAIL***: Factor-related apoptosis-inducing ligand; ***TRXR2***: Thioredoxin reductase-2*; **UCP-2***: Uncoupling Protein 2; ***xCT***: Cystine/glutamate transporter; **ΔΨm**: Mitochondrial membrane potential.

**Table 3 biomedicines-11-00693-t003:** Enyzme’s precursors and corresponding metabolic pathways targeted by the oncogenic and tumor suppressor miRNAs.

MiRNAs	Enzyme Precursor	Metabolic Pathway/Cell Compartment	Reference
let-7a	*ND4*	OXPHOS/mitochondria	[55]
let-7a	*AASDHPPT* *FASN* *G6PD* *HMOX1* *IMPDH2* *SCD*	Pantothenate and CoA biosynthesis/cytosol Fatty acid biosynthesis/cytosol, mitochondria and nucleus Transport reactions/cytosol, extracellular and nucleus Pentose phosphate pathway/cytosol and endoplasmic reticulum Heme degradation/cytosol Porphyrin metabolism/cytosol Purine metabolism/cytosol Fatty acid desaturation/cytosol Fatty acid biosynthesis/cytosol and nucleus Fatty acid desaturation/cytosol Transport reactions—cytosol and nucleus	[44]
miR-1	*ATP6* *COX1* *GPD2* *ND1* *ND4*	OXPHOS/mitochondria OXPHOS/mitochondria Glycerophospholipid metabolism/cytosol and mitochondria Acylglycerides metabolism/cytosol and mitochondria Glycolysis/Gluconeogenesis/cytosol and mitochondria OXPHOS/ mitochondria OXPHOS/ mitochondria	[56]
miR-17*	*GPX2* *MnSOD* *TRXR2*	Arachidonic acid metabolism/cytosol, endoplasmic reticulum, extracellular, mitochondria, and peroxisome Glutathione metabolism/cytosol, extracellular and mitochondria Linoleate metabolism/cytosol and endoplasmic reticulum ROS detoxification/cytosol and mitochondria Metabolism of amino acids/cytosol Pyrimidine metabolism/cytosol and mitochondria Cysteine and methionine metabolism/cytosol and mitochondria Nucleotide metabolism/cytosol and mitochondria	[73]
miR-17-3p	*GPX2* *MnSOD* *TRXR2*	Same as miR-17*	[74]
miR-27a	*CTH* *SLC7A11*	Cysteine and methionine metabolism/cytosol Metabolism of other amino acids/cytosol Transport reactions/cytosol and extracellular	[45]
miR-98	*LASS2*	Sphingolipid metabolism/cytosol	[50]
miR-133a	*UCP-2*	Transport reactions/cytosol and mitochondria	[48]
miR-140-5p	*GLUT1*	Glycosphingolipid metabolism/cytosol, endoplasmic reticulum, and golgi apparatus	[65]
miR-155	*PIK3R1* *p85α*	Inositol phosphate metabolism/cytosol p85Inositol phosphate metabolism/cytosol	[66]
miR-181a-5p	*mt-CO2* *mt-CYB*	OXPHOS/mitochondria	[68]
miR-181c	*mt-CO1*	OXPHOS/mitochondria	[62]
miR-195	*ACACA* *CYP27B1* *HMGCR*	Biotin metabolism/cytosol and nucleus Transport reactions/cytosol and nucleus Vitamin D metabolism/cytosol and mitochondria Cholesterol metabolism/cytosol, endoplasmic reticulum and peroxisome Cholesterol biosynthesis 1 (Bloch pathway)/cytosol Transport reactions/cytosol, endoplasmic reticulum and peroxisome	[70]
miR-210	*COX10*	Histidine metabolism/cytosol	[52]
miR-210-3p	*GPD1L* *HIF-1α activity*	Glycerophospholipid metabolism/cytosol and peroxisome Transport reactions/cytosol and peroxisome	[67]
miR-1291	*CPT1C*	Fatty acid oxidation/cytosol and peroxisome Carnitine shuttle/cytosol, endoplasmic reticulum and peroxisome	[72]
miR-4485	*16S rRNA*	Phenylalanine, tyrosine and tryptophan biosynthesis/cytosol	[60]

Legend. 16S rRNA: 16S ribossomal RNA; *AASDHPPT*: 4-Phosphopantetheinyl transferase; *ACACA*: Acetyl-CoA carboxylase; *ATP6*: ATP Synthase Membrane Subunit 6; *COX1*: Cytochrome C oxidase subunit 1; *COX10*: Cytochrome C oxidase subunit 10; *CPT1C*: Carnitine palmitoyltransferase 1C; *CTH*: Cystathionine gamma-lyase; *CYP27B1*: subfamily B, polypeptide 1; *FASN*: fatty acid synthase; *G6PD*: Glucose-6-phosphate dehydrogenase; *GLUT 1*: Glucose transporter 1; *GPD1L*: Dehydrogenase 1-like; *GPD2*: Glycerol-3-phosphate dehydrogenase 2; *GPX2*: Glutathione peroxidase-2; *HIF-1α*: Hypoxia-inducible factor 1-alpha; *HMGCR*: 3-Hydroxy-3-Methyl glutaryl CoA reductase; *HMOX1*: Heme oxygenase 1; *IMPDH2*: De novo guanine nucleotide biosynthesis; *LASS2*: LAG1 longevity assurance homolog 2; *MnSOD*: Manganese superoxide dismutase; *mt-CO1*: Mitochondria-encoded cytochrome c oxidase subunit-1; *mt-CO2*: Mitochondrially Encoded Cytochrome C Oxidase II mt-CYB; *mt-CYB*: Mitochondrially Encoded Cytochrome B; *ND1*: NADH dehydrogenase 1; *ND4*: NADH dehydrogenase 4; OXPHOS: Oxidative phosphorylation; *PIK3R1*: phosphoinositide-3-kinase regulatory subunit alpha; p85α: Modular protein binds and inhibits the enzymatic activity of class IA PI3K catalytic subunits; *SCD*: Stearoyl-CoA desaturase; *SLC7A11*: Solute carrier family 7 member 11; *TRXR2*: Thioredoxin reductase-2; ***UCP-2***: Uncoupling Protein 2.

**Table 4 biomedicines-11-00693-t004:** MiRNAs with oncogenic and suppressive function mediating cytotoxic response to common chemotherapeutic agents in the distinct types of cancer cells (presented by cancer type).

MiRNA	Function	Cancer Type	Mechanisms of Action/Cytotoxicity Response	Reference
miR-98	OncomiR	Bladder cancer	↑miR-98 ↑ resistance to cisplatin and doxorubicin in T24 bladder cancer cells	[50]
miR-27a	Suppressor	Breast cancer	↑miR-27a ↑ROS ↑cytotoxicity to doxorubicin and paclitaxel in MCF-7 and MDA-MB-231 cells	[45]
miR-27a	OncomiR	Breast cancer	↑ miR-27a ↑resistance to cisplatin in T-47D cells	[51]
miR-125b	Suppressor	Breast cancer	↑miR-125b ↑ROS ↑ cytotoxicity to doxorubicin in MCF-7/R cells	[46]
miR-125b	Suppressor	Breast cancer	↓miR-125b ↓cytotoxicity to doxorubicin in MCF-7 cells	[47]
miR-133a	Suppressor	Breast cancer	↓miR-133a ↑UCP-2 ↑ doxorubicin-resistant to doxorubicin in MCF-7/DOX cells	[48]
miR-223	Suppressor	Breast cancer	↑ miR-223 ↑cytotoxicity to doxorubicin or cisplatin in MDA-MB-231 cells	[75]
miR-519d	Suppressor	Breast cancer	↑ miR-519d ↑ cytotoxicity to cisplatin in T-47D-cancer stem cells.	[76]
miR-let-7a	Suppressor	Breast cancer Melanoma	↑miR-let-7a ↑Cytotoxicity to doxorubicin in MDA-MB-231 cells	[44]
miR-593-5p	OncomiR	Tongue cancer	↑miR-593-5p ↓ MFF ↑ resistance to cisplatin in Cal-27 and SCC-9 cells	[53]
miR-2392	OncomiR	Tongue cancer	↑miR-2392 ↑ glycolysis ↑ resistance to cisplatin in CAL-27 and SCC-9 cells	[38]

Legend. “↑”: enhanced; “↓”: reduced; MFF: Mitochondrial fission factor; ROS: reactive oxygen species; UCP-2: Uncoupling Protein 2.

## Data Availability

Not applicable.

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
