# Peer review of "MiRNAs Action and Impact on Mitochondria Function, Metabolic Reprogramming and Chemoresistance of Cancer Cells: A Systematic Review"

_biomedicines, 2023, doi:10.3390/biomedicines11030693_

Round 1

Reviewer 1 Report

The review of Rosolen et al is devoted to the role of miRNAs in mitochondria metabolism, metabolic rewiring and resistance against chemotherapeutic interventions. The review is well-organized; clearly describes the oncogenic and oncosupressive modes of miRNA’s action as well as miRNA’s impact on tumor chemoresistance.

To my opinion, the only minor points are:

1)    Introduction: the authors describe only several features of cancer-related metabolic rewiring; however, besides the increased glucose uptake and Warburg effect, the metabolic plasticity frequently includes intensified one-carbon metabolism, beta-oxidation of fatty acids, pentose-phosphate pathway, re-acquired ability to de novo biosynthesis of fatty acids, altered metabolism of glutamine, TCA, production of oncometabolites, etc.

It would be good to say a few words about all of these main features, if the review is devoted to metabolic reprogramming.

2)    The authors have summarized the information about miRNAs which target transcripts of different enzymes of various metabolic pathways (SCD, FASN, HMGCR – biosynthesis of fatty acids and cholesterol; G6PD – the key enzyme of PPP; CPT1 – the rate-limiting stage of fatty acids beta-oxidation; IMPDH2 – one of the most important enzymes of biosynthesis of nucleotides (purines), etc. I suggest to add one more figure which will briefly summarize and unite different metabolic pathways and miRNAs which target the certain precursors of enzymes. Or it could be the Table which will summarize what enzyme’s precursors in the certain metabolic pathway is targeted by the cognate miRNA

Reviewer 2 Report

Introduction

MiRNA should be microRNA (MiRNA), for the first appearance in the text.

Please cite this reference and discuss with the results.

Zhang, G. Q., Wang, S. Q., Chen, Y., Fu, L. Y., Xu, Y. N., Li, L., Tao, L., & Shen, X. C. (2021). MicroRNAs Regulating Mitochondrial Function in Cardiac Diseases. Frontiers in pharmacology, 12, 663322. https://doi.org/10.3389/fphar.2021.663322
